# An Overview of the Dry Eye Disease in Sjögren’s Syndrome Using Our Current Molecular Understanding

**DOI:** 10.3390/ijms24021580

**Published:** 2023-01-13

**Authors:** Kevin Y. Wu, Merve Kulbay, Cristina Tanasescu, Belinda Jiao, Bich H. Nguyen, Simon D. Tran

**Affiliations:** 1Department of Surgery, Division of Ophthalmology, University of Sherbrooke, Sherbrooke, QC J1G 2E8, Canada; 2Faculty of Medicine, University of Montreal, Montreal, QC H3T 1J4, Canada; 3School of Optometry, University of Montreal, Montreal, QC H3T 1P1, Canada; 4Department of Medicine, Division of Internal Medicine, University of Sherbrooke, Sherbrooke, QC J1G 2E8, Canada; 5CHU Sainte Justine Hospital, Montreal, QC H3T 1C5, Canada; 6Faculty of Dental Medicine and Oral Health Sciences, McGill University, Montreal, QC H3A 1G1, Canada

**Keywords:** Sjögren syndrome, dry eye, ocular surface, basic research, pathophysiology, immune pathway, diagnosis, biomarkers, proteomics, exosomes

## Abstract

Sjögren’s syndrome is a chronic and insidious auto-immune disease characterized by lymphocyte infiltration of exocrine glands. The patients typically present with ocular surface diseases related to dry eye and other systemic manifestations. However, due to the high prevalence of dry eye disease and the lack of objective and clinically reliable diagnostic tools, discriminating Sjögren’s syndrome dry eye (SSDE) from non-Sjögren’s syndrome dry eye (NSSDE) remains a challenge for clinicians. Diagnosing SS is important to improve the quality of life of patients through timely referral for systemic workups, as SS is associated with serious systemic complications such as lymphoma and other autoimmune diseases. The purpose of this article is to describe the current molecular understanding of Sjögren’s syndrome and its implications for novel diagnostic modalities on the horizon. A literature review of the pre-clinical and clinical studies published between 2016 and 2022 was conducted. The SSDE pathophysiology and immunology pathways have become better understood in recent years. Novel diagnostic modalities, such as tear and saliva proteomics as well as exosomal biomarkers, provide hope on the horizon.

## 1. Introduction

Sjögren’s syndrome (SS) is a chronic and multisystemic disease characterized by lymphocyte infiltration of exocrine glands, typically the lacrimal and salivary glands, resulting in the classical manifestations of dry eyes and dry mouth [1]. This disease was first described in 1933 by the Swedish ophthalmologist Dr. Henrik Sjögren, who observed a series of patients with xerophthalmia, xerostomia, and arthralgia.

### 1.1. Classification of Sjögren’s Syndrome

The SS is classified as primary Sjögren syndrome (pSS) when clinical manifestations occur in the absence of another auto-immune disease or as secondary Sjögren syndrome (sSS) when associated with another well-defined auto-immune disease. The most common associated diseases are rheumatoid arthritis and systemic lupus erythematosus. Other diseases include Raynaud’s disease, scleroderma, primary biliary cholangitis, and auto-immune hepatitis [2]. In this study, we address only pSS.

### 1.2. Ophthalmic Manifestations of Sjögren’s Syndrome

The symptoms of ocular surface dysfunction include foreign body sensations, burning sensations, eyestrain, photophobia, blurry vision, and red eyes. In addition, the symptoms are aggravated by prolonged visual effort (e.g., reading, screen time) and by environmental extremes (e.g., low humidity, extreme cold). However, patients may be asymptomatic or mildly symptomatic despite signs of significant ocular inflammation [3].

The findings on physical examinations include conjunctival hyperemia, decreased meniscal height (<0.3 mm), and tear break-up time (<10 s). In addition, ocular staining may reveal devitalized epithelium on the conjunctiva with the use of rose Bengal or lissamine green, and punctate epithelial erosions on the cornea using fluoresceine. Due to the decreased corneal sensation, the severity of symptoms does not necessarily correlate with the severity of SSDE. In fact, signs of severe SSDE can be found in asymptomatic patients. In advanced DED, debris such as discharge, filaments, and mucous plaques on the ocular surface resulting from dead epithelium can be revealed on a slit-lamp examination, as well as corneal calcification and keratinization of the cornea and conjunctiva. Subsequently, meibomian gland dysfunction with palpebral signs (e.g., telangiectasias and meibomian gland obstruction and atrophy) may be observed, even though SSDE initially originates from aqueous tear deficiency (ATD) as opposed to an evaporative mechanism led by lipid layer deficiency [3]. Generally, the SSDE is less symptomatic but shows more severe clinical signs than the NSSDE. Further, blurred vision tends to be worse in SSDE than in NSSDE. However, there is not a distinguishing feature to differentiate SSDE from NSSDE at the bedside [4].

### 1.3. Glandular Manifestation

The extraocular glandular manifestations present as xerostomia, a hallmark sign of SS. The oral symptoms include difficulty eating, swallowing, or speaking. The patient may develop dental caries, oral candidiasis, and chronic esophagitis due to hyposalivation [5]. Enlargement of the salivary glands is observed. This is usually due to sialadenitis secondary to lymphocytic infiltration. However, malignancy (i.e., mucosa-associated lymphoid tissue (MALT) lymphoma and primary neoplasia of the salivary gland) must be eliminated in the presence of a unilateral painless firm nodule, facial nerve palsy, and the presence of constitutional symptoms [6]. The enlargement of lacrimal glands may occur in SS but is rare as opposed to salivary glands. In the presence of enlarged lacrimal glands, investigations to rule out lymphoma, sarcoidosis, amyloidosis, IgG4-related disease, and other infiltrative diseases must be performed, and dacryoadenitis secondary to SS remains a diagnosis of exclusion [6].

### 1.4. Extra-Glandular/Systemic Manifestations

The most common extraglandular manifestation in SS is arthralgia. A total of about 10% of patients have cutaneous lesions from vasculitis and xeroderma. The pulmonary involvement beyond the sicca complex manifests as bronchiolitis and interstitial lung diseases [7]. However, of clinical relevance is the involvement of the renal system, manifesting as tubulointerstitial nephritis, and of the gastrointestinal system, affecting the liver and the pancreas [3,8]. The patients may also develop sensory neuropathy, myelitis, and meningitis. A wide range of hematological manifestations such as anemia, leucopenia, thrombocytopenia, and hypergammaglobulinemia are the main causes of mortality in patients suffering from SS. A greater risk for Hashimoto’s thyroiditis, cardiovascular disease, and depression is present [3]. The SS is the autoimmune disease most often associated with lymphoma, representing a relative risk of 13.8 for non-Hodgkin’s lymphoma. A relative risk of 1.5 for all malignant tumors and of 2.6 for thyroid tumors exists for all SS patients [9,10,11].

## 2. Pathogenesis

The pathogenesis of pSS is complex, and although no clear etiologies have been identified until today, multiple studies have reported a multifactorial pathogenesis. In the following sections, we will review the pathogenesis of pSS.

The pathogenesis can be divided into four subsequent steps (Figure 1):Interaction between genetic susceptibility and environmental exposureDevelopment of autoimmunityDestruction of lacrymal glands leads to aqueous tear deficiency (ATD)Affectation of functional lacrimal unit, with the initiation of DED vicious cycle

Each of the steps is further detailed in this section.

### 2.1. Genetic Predisposition

A crucial role for genetic predisposition in the development of pSS has been widely underscored in the literature. The genes within and outside the major histocompatibility complex (MHC) have been identified and linked to an increased risk of developing pSS. A meta-analysis, regrouping 23 patients from various ethnicities, showed a greater association between HLA class II alleles DQB1*02:01, DQA1*05:01, and DRB1*03:01 and the incidence of pSS. One of the hallmarks of pSS diagnosis relies on the presence of autoantibodies, such as anti-Ro/SSA and/or anti-La/SSB antibodies. It was shown that the presence of anti-Ro and/or anti-La autoantibodies was strongly linked to patients with DRB1*03 and DQB1*02 alleles or heterozygous DQw1 and DQw2 [12,13].

The non-HLA genes that have genome-wide association (GWA) with pSS include IRF5, TNIP1, BLK, STAT4, IL12A, and CXCR5 genes. In addition, GTF2I is a gene that helps the transcription of the immunoglobulin heavy chain and is found to be a predisposing gene in developing pSS in Han Chinese. Other genes with suggestive significance have also been identified in GWAS studies. The epigenetic factors that may influence gene expression involve DNA methylation, histone acetylation, non-coding RNAs, and gene recombination [14,15].

### 2.2. Environmental Triggers

It is suggested that some viral infections that affect epithelial cells in exocrine glands may trigger pSS disease activation in genetically predisposed patients. The Ebstein-Barr virus (EBV) is a human herpesvirus-4 (HH-4) associated with infectious mononucleosis, neoplastic disorders such as lymphoma, and other autoimmune diseases. It often infects and lies latent in the salivary glands [16]. The EBV induces autoimmunity in the pSS through multiple proposed mechanisms, including molecular mimicry and cross reactivity between the viral EBNA-2 protein and Ro-60 antigen, as well as EBER-1 and EBER-2 viral proteins and La antigens, that trigger autoreactive B or T cells [17]. It was shown that EBV can be present in ectopic lymphoid structures on glandular histopathology, and the contained plasma cells can produce autoantibodies associated with pSS. Increased prevalence and titers of serologic EBV were also found in patients with pSS when compared to healthy individuals. These findings support the role of active EBV infection in promoting lymphocytic autoreactivity in pSS. Other than EBV, clinical features of SS are associated with other viral agents such as hepatitis C virus (HCV), human immunodeficiency virus (HIV), human T-lymphotropic virus type 1 (HTLV-1) and coxsackievirus [18,19]. This autoreactivity is of significant importance in activating epithelial cells present in exocrine glands, the conjunctiva, and possibly meibomian glands. The ensuing immune and inflammatory responses further perpetuate epithelial cell activation and create a vicious cycle that favors autoimmunity [20,21].

In vivo models described throughout the literature have described the molecular pathways involved in pSS initiation by using toll-like receptor (TLR) agonists, which mimic the mechanism of action of viral agents. In addition, several studies have demonstrated that intraperitoneal polyinosinic: polycytidylic acid (poly (I:C)), a TLR3 agonist, increases type 1 interferon (IFN) expression in salivary glands and induces sialadenitis similar to SS [22,23]. Genetically modified C57BL/6 strain mice have been shown to demonstrate SS-like dacryoadenitis following poly (I:C) administration [24]. Furthermore, Hu and colleagues have shown that poly (I:C)-induced respiratory viral infection in NOD/ShiLtJ mice increased IL-33 expression in salivary glands and induced an upregulation of cytokines and chemokines, exacerbating exocrine dysfunction [25]. Other mouse model studies demonstrated SS-like disease exacerbation using administration of murine CMV or hepatitis Delta virus (HDV) and double-stranded DNA produced by viruses [23,26,27].

Additionally, there are other potential triggers for the development of Sjögren’s syndrome, including hormonal changes (e.g., menopause), and stressful events (e.g., stressful life events, major surgery, or trauma) [28].

### 2.3. Immune Response—Development of Autoimmunity

The immune response of the SS can be mainly divided into two pathways, involving the innate immune response or the adaptive immune system. In this section, we will thoroughly review the molecular mechanisms involved in each pathway, as well as the most important molecular partners involved in disease initiation, progression, and histopathological changes (Figure 2).

#### 2.3.1. Alterations in the Innate Immune Response

Innate immunity is the rapid initial response to invasion by foreign microorganisms through the recognition of pathogen-associated molecular patterns (PAMPs). It is most likely involved in pSS disease initiation and progression. This process is often recognized as autoimmune epithelitis [29]. The latter effect of innate immunity is to induce initial dysregulation of glandular epithelial cells through the release of chemokines and adhesion molecules following the activation of the IFN-related pathways [30,31].

As mentioned earlier, environmental triggers play an important role in the initiation of pSS. The viruses express PAMPs on their cell surfaces that allow their recognition by cells of the innate immune system. The PAMPs bind to pattern recognition receptors (PRRs) that are expressed on many innate immune cells as well as on professional antigen-presenting cells (APCs), such as macrophages, dendritic cells (DCs), and B lymphocytes. TLRs, a subclass of transmembrane PRR, were shown to be overexpressed in serum and exocrine glands, in both humans and animal models with SS [32]. This overexpression of TLRs can thus be linked to a higher activation of the immune system. Once activated by PAMPs, the TLR transduction signal leads to an increase in type 1 IFN activity, which engages the innate immune system [33]. Further, multiple studies throughout the literature have reported the importance of IFNs in the pathogenesis of SS.

Three types of IFNs exist: type I, type II, or type III. Type 1 IFNs are polypeptides that generally serve to limit pathogenic spread, modulate innate immunity, restrain pro-inflammatory pathways, and promote adaptive immunity. The IFNα subtypes and IFNβ remain the most studied type 1 IFNs in rheumatic diseases [34]. In SS, the principal actors involved in IFNα production are plasmacytoid DCs (pDCs). It was shown that pDCs circulating in the peripheral blood of patients with SS overexpress the marker CD40, a cell-activating marker. Furthermore, genomic analyses of peripheral blood monocytes demonstrated that more than 50% of upregulated genes were IFN-inducible genes [35]. Type 1 IFN inducible genes mainly involve the IFI44L, IFI44, IFIT3, LY6E, and MX1 genes, also known as the “IFN type 1 signature genes.” It was shown that an upregulation in the expression of type 1 IFN inducible genes was associated with an increase in pSS, greater disease activity, and higher autoantibody levels, such as anti-Ro52, anti-Ro60, and anti-La, along with higher B cell activating factor (BAFF) gene expression levels in monocytes [36]. The IFNs are mainly involved in the activation of the JAK-STAT pathway, which regulates cytokines such as IL-6, IL-7, IL-21, and IL-23. The pro-inflammatory cytokine IL-6 was shown to be increased in the biological fluids of pSS patients and mediate its pathogenesis through the modulation of B and T cell differentiation and activation. Similarly, IL-21 and IL-23 are involved in the activation of B cells as well as in the generation of IL-17-secreting T cells (Th17 cells) [37]. IL-23 is also involved in the expansion of innate lymphoid cells (ILC), which mainly produce IL-17, thus enhancing the vicious cycle of inflammation. It is clear that cytokines are an important bridge to the adaptative immune response: IFN-gamma production was shown to induce chemokines CXCL9, CXCL10, and CXCL12, as well as the expression of their receptor CXCR3 on T-cells [38]. This step was shown to be crucial in T-cell recruitment and infiltration in the salivary glands of patients with SS. Recently, it was shown that the use of baricitinib, a JAK-STAT transduction inhibitor, inhibits IFN- gamma -induced CXCL10 production in human salivary gland ductal cells, further confirming the role of IFNs in disease progression. Moreover, there are reports of patients receiving IFN type 1 treatment for HCV who have developed SS-like symptoms, which further suggest the pathogenic role of an upregulated IFN pathway in pSS [39,40].

A key partner in SS pathogenesis are epithelial cells themselves; epithelial cells in exocrine glands are directly involved in enhancing the innate immune system in pSS while also acting as a target for autoimmunity. The salivary gland epithelial cells secrete costimulatory molecules (e.g., CD86), cytokines (e.g., IL-21), and chemokines (e.g., CXCL12) that promote T-cell interaction, B-cell activity, and leukocyte recruitment [41,42]. The pDC are a subtype of DCs that are engaged by self-antigens through TLRs, mainly TLR-7 and TLR-9, but also TLR-2 and TLR-4, leading to type 1 IFN and other pro-inflammatory cytokine secretion [43]. It was shown that pDCs in the peripheral blood of pSS patients are decreased, and they are linked with enhanced glandular inflammation, suggesting abnormal apoptosis activity. The number of pDCs in salivary glands in pSS patients is correlated with the level of cells that secrete IFN-α [35,44].

In addition, among the cytokines upregulated by IFNs, BAFF has been gaining increased attention for its role in the pathogenesis of SS. The exaggerated responses of the type 1 IFN pathway promote B cell activation and autoantibody proliferation by stimulating the production of BAFF [36]. The BAFFs were shown to be found at higher levels in the salivary glands and serum of patients with pSS. Its expression was shown to correlate with autoantibody (anti-Ro/SSA and anti-La/SSB) levels and with a more severe disease phenotype [45,46,47]. The high BAFF levels prolong autoreactive B cell survival and may even favor lymphoproliferative disease through continuous stimulation of B cells [48]. Moreover, it was shown that BAFF transgenic mice develop exocrine inflammation and hyposalivation like SS patients. The BAFF enhances B-cell activation, and is thought to be the crossroad linking innate to acquired immunity [33,49].

#### 2.3.2. Alterations in the Adaptative Immune Response

The BAFF activation promotes plasma cell survival and hypergammaglobulinemia (hallmarks of B-cell dysregulation), therefore increasing the quantity of plasma cells that contain autoreactive cells. As mentioned earlier, B-cells are attracted to salivary gland epithelial cells (SGEC) by the action of enhanced CXCL3 and CXCL4 expression, which causes cell infiltration and glandular tissue inflammation [50]. The CXCL3 is also involved in the accumulation of T follicular helper (Tfh) cells in the exocrine glands, and in combination with B cells, they contribute to the formation of ectopic germinal center (GC)-like structures [51]. It was shown that the GC-like structures positively correlated with the focus scores, cell infiltration status, and autoantibody levels [52]. Once recruited to the salivary glands and GC-like structures, B-cell activation and plasma cell differentiation are promoted through the inflammatory microenvironment. It was shown that type 1 IFNs not only activate BAFF but also contribute to higher levels of proliferation-inducing ligand (APRIL), which are both crucial markers in SS pathogenesis [53]. The stimulation of the B-cell receptor (BCR), binding of the CD40 receptor to its ligand, CD40L, and cytokine production in GC-like structures allow the generation of isotype-switched cells, therefore contributing to B-cell hyperreactivity and autoantibody generation [41].

T lymphocytes also play important roles in pSS disease pathogenesis. CD4+ T cells can differentiate into Th1 and Th2 subtypes. The Th1 subtype promotes the production of IFN-γ and IL-12. It was shown in non-obese diabetic (NOD) mice lacking IFN-γ expression that the initiation and presentation of SS-like diseases were prevented [54]. Overexpressing IL-12, an essential IFN-γ inducer, favors pSS-like symptoms in mice. Moreover, the IFN signature, as previously described in pSS patients, seems to be closely linked to IFN-γ in many patients. This highlights the major role of the IL-12 cytokine in pSS pathogenesis. The Th1 and Th2 may be present in different stages of pSS disease, but this process is not fully understood. However, the Th17 cells seem to be involved in the disease process, as they produce cytokines including IL-17A through IL-17F along with TNF-α and IL-22. These cytokines have been observed in the blood and saliva of patients with SS. Moreover, IL-17 expression was related to the severity of disease as well as maintaining inflammation in pSS. The treg cells, downregulators of effector T cell activation, may be involved to some extent in pSS pathogenesis, but studies have been controversial, and their role is not yet fully elucidated [33,55,56,57,58].

In following sustained inflammation through the innate immune system and GC-like structures, the SGEC activate their apoptotic pathway. The epithelial cells release ribonucleoprotein complexes (Ro/SSA and La/SSB) that trigger autoimmunity by recruiting DCs within exocrine glands [58,59,60]. In a SS mouse model, it was shown that epithelial cell apoptosis was essential in promoting inflammation. However, focal inflammatory response and secretion of cytokines (such as IFN-γ and TNF-α) disrupt tight junction structure and dysregulate local exocrine gland secretory function, which may contribute to the clinical signs of sicca and inflammation in pSS patients. This inflammatory microenvironment further activates epithelial cells, and the vicious cycle of inflammation increases autoimmune activity in pSS, leading to chronic inflammation [20,21]. The role of autoimmune antibodies in pSS is among the key steps involved in adaptative immunity. The autoantibodies anti-Ro/SSA and anti-La/SSB are present, respectively, in approximately 50–70% and 25–40% of patients with pSS, and represent the hallmark of this disease. They can be detected up to 18–20 years before clinical manifestations of pSS, as demonstrated by pre-symptomatic serum samples from pSS patients, suggesting that the auto-immune and pathogenic processes may long precede clinical evidence of disease. Further, they do not seem to contribute directly to pSS pathogenesis, but secretion of these antibodies locally may promote immune complexes that could potentially activate IFN¬α production. Recently, studies have been suggesting two distinct entities in anti-Ro/SSA autoantibodies, which are anti-Ro52 and anti-Ro60 antibodies [61,62]. On the basis of firsthand observation, the autoantigen Ro52 was shown to act as an E3 ligase, by inhibiting the NFκB signaling pathway and the Bcl-2 anti-apoptotic protein, therefore having anti-proliferative and pro-apoptotic properties. On the other hand, the autoantigen Ro60 is known to bind to non-coding small RNAs (i.e., YRNA) and regulate type 1 IFN gene expression. Finally, the autoantigen La/SSB was shown to be involved in microRNA expression regulation [63,64,65,66].

### 2.4. Destruction of Lacrymal Glands and Aqueous Tear Deficiency

Autoimmunity initiates infiltration and destruction of lacrimal glands, associated with the histopathological hallmark of SS: lymphocytic infiltration that mainly surrounds striated ducts of exocrine glands. This distribution leads to periductal foci. The focus score currently used by the 2016 ACR/EULAR diagnostic criteria is defined as the number of periductal foci in a 4-mm^2^ area. The ductal and glandular atrophy are also present [67]. A microscopic view of a majority of T cell infiltrates, mainly CD4+ but also CD8+, generally signifies less severe disease and inflammation. A more important infiltration of exocrine glands causing tissue architecture destruction involves a predominance of lymphocyte B cells [51]. Eventually, the lymphocytic infiltration may organize into tertiary lymphoid tissue with a germinal center (GC) that is similar to secondary lymphoid tissues. These ectopic GC-like structures can be present in 18–59% of SS patients. The disruption of the architectural integrity of exocrine glands by the formation of GC can lead to hyposecretion, as seen in pSS, by perpetuating local inflammation and autoantibody formation [40,56,57,58,59]. Corsiero and his colleagues have provided data supporting defects in B-cell tolerance checkpoints in pSS patients, which leads to the accumulation of autoreactive naïve and memory B cells. These GC-like structures increase the risk of lymphoproliferative diseases through continuous B-cell activation and clonal expansion [68]. In a study of 175 pSS patients where labial SG tissue biopsies were performed, 86% of patients who developed non-Hodgkins lymphoma (NHL) had GC-like structures versus 22% for patients without NHL [69]. The pSS salivary gland biopsies demonstrated that around 45–50% of CD4+ T lymphocytes, 20% of CD8+ T lymphocytes, and 20% of B cells form these structures [29].

The lymphocytic infiltration and destruction of lacrymal glands eventually lead to aqueous tear deficiency and, subsequently, hyperosmolarity of the ocular surface’s tear film. The tear film hyperosmolarity activates epithelial inflammatory cascades (i.e., MAPK+ et NFκB+ pathways), which increase pro-inflammatory cytokines. Interestingly, several of these cytokines are specific to SS (IL-1ra, IL-2, IFN-γ, IP-10) offering the potential for the development of disease markers [70].

### 2.5. Affectation of Functional Lacrimal Unit and Vicious Cycle of SSDE

The pro-inflammatory cytokines caused by ocular surface hyperosmolarity affect substantially the entire functional lacrimal unit. It is noteworthy that all the constituents of the functional lacrimal unit, as well as all three major components of the tear film, are affected at this stage [71,72]: Meibomian gland dysfunction and atrophy reduce the lipid layer of the tear film, leading to increased evaporation.Apoptosis of goblet cells decreases the mucin layer of the tear film, leading to decreased wettability.Degeneration of epithelial cells leads to a loss of microvilli, also contributing to decreased wettability.Neurogenic inflammation of the lacrimal gland further contributes to aqueous tear deficiency.Disturbances of corneal nerves lead to decreased corneal sensation and blinking reflex.

These events lead to an unstable ocular surface tear film, which contributes back to ocular surface hyperosmolarity, maintaining the vicious cycle of chronic ocular surface inflammation [70]. Despite SSDE starting with an aqueous tear deficiency, the clinical manifestations of SSDE are similar to those of NSSDE, as the entire functional lacrimal unit eventually gets compromised. This makes the clinical distinction between the two difficult.

## 3. Current Diagnostic Challenges Encountered by Clinicians

SS affects 0.06 percent of the world’s population. Women make up over 90% of those affected. It has been estimated that 10% of patients with dry eye disease (DED) suffer from SS. However, two-thirds of these patients remain undiagnosed, and a median diagnostic delay of 10 years is reported [73]. Several factors may explain the underdiagnosis of SS:Dry eye and dry mouth are highly prevalent, making it difficult for clinicians to identify patients with underlying SS [73].SS has a broad spectrum of non-specific clinical manifestation, as well as an insidious onset [74].Patients with SSDE may be asymptomatic or mildly symptomatic despite signs of significant ocular inflammation [74].Reliable and effective screening tools and algorithms to determine which DED patients should be worked up for SS are lacking [4].Ophthalmologists underestimate the importance of SS and consequently refer few patients with DED for SS workup [75].

### 3.1. Importance of SS Diagnosis

Untreated SSDE may result in vision-threatening complications such as neurotrophic keratitis, resulting in corneal thinning, ulceration, melting, and perforation [3,76]. Moreover, SS-related type III hypersensitivity reactions can lead to immune-complex depositions resulting in severe inflammation of other ocular structures such as scleritis [77], uveitis [78], optic neuropathy [79], and retinal vasculitis [71,80].

Sjögren’s syndrome tends to have a relatively mild and benign course. [28] However, around a third of patients with SS will develop systemic complications, contributing to a diminished quality of life and increased morbidity and mortality. The SS is associated with an alarmingly high risk of lymphoma development, which is a particularly concerning complication among the various systemic complications that may arise [3,73]. Early detection of SS may prevent complications and allow for clinical surveillance for the development of serious systemic manifestations. The early recognition of SSDE can also influence future ocular surgical decision-making, as refractive surgery is an absolute contraindication [81,82] and blepharoplasty is a relative contraindication in the presence of SSDE [83].

The ESSDAI (European Sjögren’s Syndrome Disease Activity Index) is a tool used to assess the disease activity and damage in individuals with Sjögren’s syndrome. It is a questionnaire that consists of several items that assess a variety of clinical features of Sjögren’s syndrome, including constitutional, glandular, articular, cutaneous, pulmonary, renal, muscular, central, and peripheral nervous systems, hematological manifestations, as well as lymphadenopathy and laboratory abnormalities. Each item is scored on a 0–3 scale, with higher scores indicating more severe disease activity. The ESSDAI is mostly used in research to assess the effectiveness of treatments for Sjögren’s syndrome [84]. However, for clinical purposes, the ESSPRI (European Sjögren’s Syndrome Patient Reported Index) is more practical for clinicians and asks the three following questions:How severe has your dryness been during the last 2 weeks?How severe has your fatigue been during the last 2 weeks?How severe has your pain (joint or muscular pain in your arms or legs) been during the last 2 weeks?

Each question is scored on a 0–10 scale, with higher scores indicating more severe functional impairment. The ESSPRI is commonly used to assess the impact of the disease on patients’ quality of life and daily functioning [84].

Other questions to ask patients to help clinicians investigate the possibility of Sjögren’s syndrome include:Do you experience a dry mouth or dry eyes?Do you have difficulty swallowing or speaking due to a dry throat?Do you experience dryness or irritation in other areas of your body, such as your nose, skin, or vagina?Do you have joint pain or swelling?Do you have fatigue or a general feeling of being unwell?Do you have a family history of autoimmune disorders?

### 3.2. Conventional Diagnostic Tools and Criteria

At present, a reliable and effective diagnostic tool is lacking. The American College of Rheumatology/European League Against Rheumatism criteria for primary Sjögren’s syndrome established in 2016 have been the most studied and used system (Figure 3) [85]. In order to calculate the diagnostic score for pSS, the following criteria should be used:Labial salivary gland with focal lymphocytic sialadenitis and a focus score of 1 or more foci per 4 mm^2^. (3 points)Presence of auto-antibodies, including anti-Ro or anti-La. (3 points)Ocular staining score of 5 or higher or a van Bijsterveld score of 4 or higher in at least one eye. (1 point)Schirmer’s test 5 mm/5 min or lower in at least one eye. (1 point)Unstimulated whole saliva flow rate of 0.1 mL/min or lower. (1 point)

Additionally, each criterion listed above is assigned a certain number of points, and the total score is calculated by adding up the points for each criterion. A patient is considered to have pSS if they meet a certain number of criteria and have a total score of 4 or more points.

This set of Sjögren’s syndrome (SS) classification criteria demonstrates a lack of appreciation for the importance of eye condition in SS, since the presence of a positive labial salivary gland biopsy and a positive SS antibody are both given three times more weight than the findings of the ocular examination (such as ocular surface staining or the Schirmer test). Moreover, even though these criteria are established diagnostic criteria, they all present difficulties in their clinical application:The dosage of serological markers such as anti-RO/SSA and anti-La/SSB antibodies, as well as ANA and rheumatoid factor (RF), are insensitive and thus cannot be used to screen for SS. It was shown that the reliability of ANA titers for SS is approximately 80% [86] and reaches a highest sensitivity value of 68.3% [87]. The prevalence of serological RF in SS patients was shown to be approximately 51% [88], with a sensitivity value of 53% [89]. As for the autoantibodies, they might be undetectable at the early stages of the disease. In fact, it was shown that the sensitivity for anti-RO/SSA and anti-La/SSB antibodies varies between 69% and 77% and 39% and 44%, respectively [90].The saliva flow rate and Schirmer’s test are rarely performed in a clinical setting since they are not specific to SS [4] and take a significant amount of time to perform in busy ophthalmology practices.Van Bijsterveld Score (VBS) and Ocular Staining Score (OSS) systems are seldom used by ophthalmologists since the rose Bengal is an irritant to the ocular surface and that the lissamine green is not available in most of the eye clinics. Moreover, a high OSS remains unspecific for SSDE and can be seen in NSSDE [4].A minor salivary gland biopsy is insensitive at the onset of the disease. Although it is generally a well-tolerated procedure, the risk of hemorrhage, infection, paresthesia, and mucocele formation is still present [91].

These challenges highlight the need for sensitive and reliable diagnostic tools for the early diagnosis of SS that present practical clinical usage at a reasonable cost.

### 3.3. Other Diagnostic Tools

#### 3.3.1. Serum Testing and Biomarkers

Other than the most commonly used biomarkers (i.e., anti-SSA, anti-SSB, ANA, and RF), other serum testing and biomarkers have recently been studied in the hope of improving the current diagnostic armamentarium. In a recent study involving 74 patients with pSS, serum vitamin D 25(OH)D_3_ was found to correlate with SS signs but not with symptoms. This was the first study to clearly establish a link between serum Vitamin D and pSS, and it involved only female participants with a mean age of 53 years old, as pSS disproportionately affects women in later stages of life. Notably, vitamin D is also an immunomodulator in other structures. A deficiency has been correlated with the development of autoimmune diseases such as SLE, rheumatoid arthritis, multiple sclerosis, and SS [92]. The lower levels have been positively correlated with the severity of DED signs such as the Schirmer I test and TBUT but have a strong negative correlation with ocular surface staining. However, no correlation appears to exist with symptoms such as those evaluated with the Ocular Surface Disease Index (OSDI). Further investigation is needed to clearly establish this serum biomarker in the diagnostic criteria for SS, but it is already tested in usual blood panels, so it might be more readily accessible if proven to be an accurate diagnostic factor.

Furthermore, small airway respiratory disease and chronic obstructive pulmonary disease are more prevalent in pSS patients, both conditions are linked by systemic inflammation. Tests for pulmonary disease include a non-invasive induced sputum analysis, in particular for B-cell activating factor (BAFF), interleukin-6 (IL-6), and interleukin-8 (IL-8) [93]. The higher levels have been found in pSS patients. The SS includes, by definition, B-cell hyperactivation, and BAFF is a known modulator of this hyperactivation. The BAFF is increased in the sputum of non-smoking patients and has been recently identified as a valid biomarker in SS by the European League Against Rheumatism (EULAR). The advantage of having this potential biomarker as a diagnosis criterion is that it can be detected by enzyme-linked immunosorbent assay (ELISA) and is thus less invasive than a salivary gland biopsy, which is one of the current diagnostic criteria.

A study in 2021 revealed how serum biomarkers can be used to differentiate between SS, rheumatoid arthritis (RA), and systemic lupus erythremous (SLE). A serum concentration of 63 markers was measured in 95 patients. The concentrations of BDNF and I-TAC/CXCL11 significantly discriminate between SS and RA. Concentrations of sCD163, Fractalkine/CX3CL1, MCP-1/CCL2, and TNFa can differentiate pSS from SLE. Furthermore, the combination of low concentrations of BDNF and Fractalkine/CX3CL1 was highly specific for pSS (specificity 96.2%; positive predictive value 80%) in comparison to RA and SLE, as was the combination of high I-TAC/CXCL11 and low sCD163 (specificity 98.1%; positive predictive value 75%). BDNF, I-TAC/CXCL11, sCD163, and Fractalkine/CX3CL1 are involved in networks of signaling pathways. These networks have proteins involved in cell-to-cell signaling interactions, cellular movement, immune cell trafficking, and the inflammatory response. Their interaction further implies the role of Nf-kb and IL-17 signaling pathways. Further, Nf-kB I expression is increased in pSS patients’ salivary gland cells. This pathway is activated in innate or acquired immunity cells, which in turn leads to the activation of other inflammatory pathways like IL-17. However, IL-17 in itself, just like TNFa, is not specific to SSDE as it is found in Fractalkine/CX3CL1, which is a chemokine involved specifically in chemotactic activity on monocytes and T-lymphocytes and the extravasation of leukocytes towards the inflammation site. The I-TAC/CXCL11 is an IFN-gamma-inducible chemokine, whose role is to attract activated T-lymphocytes, natural killer cells, and macrophages to the inflammation site [94].

#### 3.3.2. Tear Film Analysis

##### Tear Film Osmolarity

According to the second international Dry Eye Workshop, DEWS II (2017), increased tear osmolarity is part of the definition of the dry eye disease, along with increased ocular surface inflammation. The tear osmolarity can distinguish between a healthy and DED eye. It is performed using a clinically approved device for in-office testing, such as TearLab ^®^, that analyzes a small sample of tears in a few minutes. A tear osmolarity threshold of 308 mmol/L most strongly suggests DED and is universally accepted. However, studies have shown lower threshold values to also strongly correlate with mild DED. For instance, one study showed a 98.4% positive predictive value for a threshold of 305 mmol/L [95]. Healthy eyes tend to have an osmolarity closer to 298 mmol/L. It does not always correlate with symptoms, as DED symptoms can be non-specific or altered by psychological effects, but it does correlate with effective therapy. Further, various factors alter tear film osmolarity, such external factors include contact lens wear or phacorefractive surgery. Systemic conditions may also affect tear film osmolarity, such as diabetes [96]. Tear film osmolarity does not help clinicians differentiate SSDE from NSSDE.

##### Tear Ferning

The tear fern is an inexpensive and simple way to distinguish between dry eye and non-dry eye patients and has shown good sensitivity and specificity in SS; however, studies may vary. The tears are collected in a capillary tube from the lower tear meniscus. A tear sample is then placed on a glass slide and dried at a controlled room temperature and humidity for accurate results. A ferning pattern can be observed in healthy eyes, which exhibit tight, dense, and uniform ferns, whereas DED tears have irregular, shorter, and more spaced ferns [97].

#### 3.3.3. Advanced Imaging

##### In Vivo Confocal Microscopy

In vivo confocal microscopy (IVCM) assesses the corneal epithelium, sub-basal nerves, stroma, and endothelium of the cornea. The corneal nerves are essential for epithelial integrity, maintaining corneal sensation, and wound healing. The meibomian glands and the lacrimal glands’ shape and integrity can also be assessed using this non-invasive technique [98]. The inflammatory process of SS ultimately also affects the corneal subbasal nerve plexus. In a 2022 retrospective study of 71 patients, which compared SS patients with a healthy control group and an MGD group, corneal sensitivity was reduced, the density of the corneal subbasal plexus was reduced, and the tortuosity of the nerves increased as well as the number of inflammatory cells. The average number of subbasal neuromas increased when compared to the control group, even more if the patients presented with small fiber neuropathy [99].

##### Meibography

Meibography is the in vivo imaging of the Meibomian glands, located posteriorly, inside the upper and lower eyelids. Images are acquired using adaptive transillumination and interferometry. They can then be compared to clinical scales, and meibomian gland integrity, shape, and loss can be measured. Meibomian gland dystrophy is a type of DED that can occur concurrently with reduced tear production that is typically seen in Sjogren syndrome. One study in 2018 by Kang et al. compared the eyes of NSSDE, SSDE, and healthy eyes and concluded that there is a greater MG impairment in SSDE than NSSDE (poorer mean meiboscore and MG expressibility) [74]. A recent study in 2021 on 108 female patient eyes showed a strong positive correlation between the duration of pSSDED and meibomian gland dropout (r = 0.776, *p* < 0.001) [100]. While meibography does not differentiate between SSDE and NSSDE, it is still a useful tool for clinicians as it enhances diagnosis accuracy.

##### MRI/CT/US Scan of the Lacrimal Gland

In certain cases, the lacrimal can be imaged to assess the shape and loss of function using magnetic resonance imaging, computerized tomography, or ultrasound. In pSS patients, salivary gland ultrasonography is currently used as a diagnostic tool and to follow-up on anatomical changes, response to therapy, and activity [101]. The MRI is also used, as MRI sialography is the gold standard for staging the disease. A recent study in 2022 on 23 pSS patients and 23 controls was the first to show that radiomics, which is the mathematical analysis of radiology images, could be used to quantitatively differentiate between the textural features of pSS and healthy lacrimal glands [102]. In pSS, both lobes of the lacrimal gland are affected. The diseased tissue produced a modified MRI signal from the healthy tissue that could be seen with a coronal T-1 weighed image with fat saturation.

## 4. On the Horizon—Novel Diagnostic Modalities

### 4.1. Tear Film’s Molecular Analysis—Tear Proteomics

The tear film analysis could help predict the possibility of a Sjogren syndrome diagnosis when ocular symptoms first arise. Various tear biomarkers, such as protein concentrations, have been studied in recent years, such as proteins concentrations. These are found to be predictive of the inflammatory response, the antibacterial reaction, and wound healing [97]. Further, the tears are a readily available bodily fluid and contain a wide array of proteins.

The human tear film contains over 1500 proteins that can combine to form over 200 peptides. Various techniques of mass spectrometry are used to analyze the proteome. Over half of the proteins are classified as intracellular or associated with the plasma membrane, and the rest are extracellular [97]. Tear proteomics can be a non-invasive and highly specific diagnostic tool for Sjogren’s syndrome. Some proteins are specific to SS, such as the gel-forming mucins MUC5AC, while lactoferrin and group IIA phospholipase A2 (PLA2G2A) can be altered in other aetiologies of dry eye disease, such as meibomian gland dysfunction.

Various mucins are produced in the tear film to lubricate the corneal and conjunctival surfaces, which is vital during blinking. They also maintain hydration and homeostasis and prevent pathogens from binding to the epithelium [97]. A reduced expression of mucin-related genes correlate with dry eye symptoms. For instance, conjunctival goblet cell-specific MUC5AC mRNA and proteins are reduced in patients with DED and are negatively correlated with increased conjunctival staining. The MUC5AC is a mucin that is gel-forming (as opposed to small-soluble or transmembrane) and has decreased levels in cases of ocular surface inflammation in various studies, alongside the gel-forming mucin MUC19 and the transmembrane mucin MUC16 [103].

Matrix metalloproteinases (MMPs) are enzymes that disrupt the epithelial surface barrier by cleaving important junctions. This leaves the epithelium more desiccated and fragile, which leads to an irregular corneal surface and higher tear osmolarity, thus increased inflammation [104]. The MMP-9 gene is upregulated in cases of ADDE, MGD, post-LASIK DE, conjunctivochalasis, and SSDE. Recently, molecular testing in ratios, instead of just evaluating individual biomarkers alone, has been found to be more specific for predicting disease. For instance, the higher proportion of patients with SS-dry eye than non-SS had lower tear TSP-1 levels (55% vs. 29%, 95% confidence interval [CI] = 1.64 to 5.35, *p*  <  0.05) and higher tear MMP-9 levels (65% vs. 24%, 95% CI = 4.46 to 19.81, *p*  <  0.05). TSP-1 is expressed by the ocular epithelial cells and has immunoregulatory properties; thus, it can inhibit MMP-9. When its expression is reduced, a corresponding DED condition can appear in human eyes. The ratio between MMP-9 and its regulatory molecule, thrombospondin (TSP-1), which is expressed by the epithelial cells, has been proven to accurately distinguish between SS dry eye and non-SS dry eye, as a lower ratio signals a SS condition [105]. However, the tear TSP-1/MMP-9 ratio is reduced in patients with SS-DED when compared to non-SS (B = −2.36, 95% CI = −3.94 to −0.0.79, *p*  <  0.05), regardless of tear MMP-9 levels. This study was the first to show that using the TSP-1/MMP-9 ratio is a better diagnostic tool than using the two values separately, which are not SS-specific [106].

The dry eye assessment and management (DREAM) study, a multi-centered study funded by the National Eye Institute, National Institute and the of Health, evaluated autoantibodies from serological samples of 494 subjects as part of their clinical trial. One autoantibody, salivary protein-1 (SP-1), was associated with underlying SS [107]. The SS patients had a higher prevalence of autoantibodies (33%), when compared with those with other immune diseases or no SS (*p* = 0.02). The parotid secretory protein and carbonic anhydrase 6 were not significantly associated with SS (*p* = 0.33 and *p* = 0.31, respectively).

Additionally, other proteins have been predictive for SSDE, in recent studies, even more than current clinical tests: LACTO, LYS-C, and LIPOC-1. They are produced in the lacrimal gland secretory granules and are thought to be expressed together. LIPOC-1 transports lipids in tears for ocular surface protection, and LACTO and LYS-C are long-chain peptides that have bacteriostatic, anti-inflammatory, and antioxidant functions. Lower levels of these three proteins are associated with SS dry eye disease. In addition, LACTO and LIPOC-1 have shown high accuracy in the diagnosis of SS, comparable with an autoantibody profile. Alongside these proteins, albumin was also measured and found to have a good predictive quality as it signals subclinical inflammation from the leakage of inflamed conjunctival vessels. In a chart review of 110 suspected SSDE patients, with 35 being diagnosed with SSDE, currently approved clinical tests showed lower diagnostic accuracy (OSDI > 44 AUC 0.57, Schirmer test ≤ 5 mm AUC 0.59, FTBUT ≤ 3 sec AUC 0.72, Jones test ≤ 4 mm AUC 0.68, corneal staining  >  2 AUC 0.51, conjunctival staining  >  2 AUC 0.78). Tear proteins showed an AUC of 0.79 for LYS-C ≤ 1.5 mg/mL, 0.94 for LACTO ≤ 20%, 0.89 for LIPOC-1 ≤ 10% and 0.79 for ALB ≥ 15% [108].

Cell autophagy is a late response to hyperosmotic stress in the inflammatory cascade that characterizes dry eye disease. It maintains homeostasis by recycling cellular components in secretory epithelial cells involved in SS. The first study to prove this particular link will be published in 2020 with an in vitro dry eye model, using fresh human corneal tissue obtained from donors within 72 h of death [109]. The autophagy-related proteins 5 and 7 (ATG5, ATG7), the LC3B protein, and the autophagic cell biomarkers, ULK1 and Beclin-1, have been identified at higher levels in tears and in epithelial conjunctival cells in SS eyes versus non-SS dry eyes. These biomarkers have shown even higher specificity than currently used ocular surface tests (ocular surface staining, TBUT, and Schirmer tests) and ocular symptom assessments (the OSDI questionnaire). They represent a potential avenue for novel SS testing that offers high specificity. Other inflammatory cytokines present in the tears, such as interleukin-6, 7, or TNF-(alpha), are found at higher levels in DE patients but not specifically in SSDE.

Another promising biomarker seems to be epidermal fatty acid binding protein (E-FABP). The tear concentration of E-FABP is lower in SS eyes than healthy eyes; however, it shows no difference in serum or salivary concentration. In a prospective case-control study of 11 patients newly diagnosed with SS with untreated SSDE, the E-FABP tear concentration was 323.5 ± 325.6 pg/mL, a statistically significant difference from the 4076 ± 5746 pg/mL (*p* = 0.0136) found in the control group of healthy individuals. It is correlated with ocular surface epithelial damage and ocular DED symptoms. The FABP protein family is already used in the diagnosis of oxidative-related diseases such as renal failure and myocardial infarction [110].

Up until now, half of the candidate biomarkers lack validation, and although they represent potential avenues for diagnostic testing for pSS, further research is needed [97]. These biomarkers could even help screen potential DED and pSS patients before symptoms arise, and they could help with the prognosis, diagnosis, and follow-up of these patients.

### 4.2. Saliva Molecular Analysis—Salivary Proteomics

The salivary proteomics levels might offer a non-invasive testing avenue to diagnose pSS. In a 2020 study aiming to differentiate between pSS and non-pSS, thus patients having symptoms but not fulfilling the diagnostic criteria, 1013 proteins were identified in whole saliva, 219 in plasma, and 3166 in salivary gland tissue. Among the hundreds of proteins in saliva, plasma, and salivary gland tissue, 40 proteins in saliva differed between the two groups but not in plasma or salivary gland tissue. The proteins involved in inflammatory processes, such as the combination of elastase, calreticulin, and tripartite motif-containing protein 29, were upregulated, whereas those involved in salivary regulation were reduced [111]. This combination yielded 97% accuracy in diagnosing pSS. The salivary proinflammatory cytokines are also more elevated in SS patients, such as IL-6 and TNF-alpha; however, these are not specific to SS alone.

Other salivary biomarkers under study include salivary B-2 microglobulin (B2m), salivary lactoferrin, salivary neutrophil gelatinase-associated lipocalin (NGAL), salivary soluble sialic acid-binding immunoglobulin-like lectin (Siglec)-5, salivary autoantibodies, salivary calprotectin, salivary carbonic anhydrase VI, and salivary adiponectin [112].

The B2m is part of the major histocompatibility complex 1, and its elevated presence in pSS patients results from lymphocyte activation and infiltration in the salivary glands. The salivary autoantibodies are a promising avenue. In addition, serum anti-Ro/La is already a diagnostic criterion for pSS, but it is also found in saliva. Further, a total of four other autoantibodies have been validated for overexpression in pSS: anti-histone, anti-transglutaminase, anti-SSA, and anti-SSB. Another autoantibody that has been proven to be elevated in pSS is the muscarinic type 3 receptor. It modulates secretion in salivary acinar cells. The AUC for anti-M3R antibodies in saliva was 0.84 and in plasma, 0.95 [112].

The salivary carbonic anhydrase VI is a metalloenzyme that, when downregulated in pSS, reduces mouth pH, worsens cavities, and increases infection risks. The salivary adiponectin is secreted by the salivary glands, which are located in adipose tissue, and, along with adenosine deaminase (ADA), is involved in the development of B and T lymphocytes and monocyte differentiation. In pSS, they have been found to correlate with IFN-gamma, IL-1, IL-8, and TNF-alpha [112].

The lactoferrin is present in secretions and has antimicrobial qualities as well as regulator properties for cytokine production in T-helper cell type 1. It is elevated in pSS but also in periodontitis and SLE, thus it is not specific to pSS. The NGAL serves in the transportation of small, hydrophobic molecules such as hormones and fatty acids but is also upregulated in infection, diabetes, cancer, and obesity, thus it is not pSS-specific. The siglec-5′s role is not clearly determined in pSS, but it is still found at elevated levels in the peripheral blood mononuclear cells of pSS patients. Finally, S100 proteins, including calprotectin (S100A8/A9), are known to increase proinflammatory cytokines and modulate the inflammatory response in various autoimmune conditions, such as pSS and RA [112].

Further studies are needed to confirm their role as clinically useful biomarkers.

### 4.3. Exosome as a Diagnostic Tool

#### 4.3.1. What Are Exosomes?

The extracellular vesicles (EVs) have a pivotal role in the maintenance of cellular homeostasis as well as in the pathogenesis of certain diseases, such as autoimmune disorders. Conventionally, the EVs can be divided into three categories: exosomes, microvesicles, and apoptotic bodies. There are two main factors that determine the classification of Evs: the EVs size and its biogenesis pathway. The exosomes are synthesized through the endocytic pathway and from the smallest EVs, followed by microvesicles and apoptotic bodies [1]. Over the last few years, multiple studies have placed an emphasis on the role of exosomes in the pathogenesis of SS (REF). More recently, studies have been proposing a role for exosomes in the diagnosis and treatment of SS (REF). Therefore, in this section, we will review the main fundamental concepts in the biogenesis of exosomes in order to better understand their clinical use in the context of SS.

#### 4.3.2. Exosome Biogenesis

The exosomes are nanosized vesicles synthesized through the endocytic pathway and released in the microenvironment by exocytosis. The endocytic pathway is involved in the biogenesis of multivesicular bodies (MVBs), which contain numerous intraluminal vesicles (ILVs) [113]. The secreted ILVs further become exosomes in the extracellular space. The endocytic pathway for exosome biogenesis can be divided into six steps (Figure 4).

The first step within the endosomal system is the internalization of biocellular markers, particles, fluids, and lipids by endocytosis in the cytosol. The internalized components form the early endosome, a focal and crucial point in the endocytic pathway. In these vesicles, numerous sorting events are initiated in order to determine the fate of internalized proteins [113]. These components can either be directed to lysosomes for degradation, to the cell membrane for recycling, to the trans-Golgi network, or transported to the MVBs by the formation of ILVs. ILV biogenesis is mainly operated by the endosomal sorting complex required for transport (ESCRT) proteins [114]. In addition, following the ubiquitination of cargoes in the early endosome, these components are sorted into PI3P enriched endosomal compartments [115]. This latter step is mediated by the hepatocyte growth factor-regulated tyrosine kinase substrate (Hrs) of the ESCRT-o complex, which recognizes ubiquitinated cargoes [116]. Once the cargoes are sorted out, ESCRT-o induces the recruitment of ESCRT-1 through a heterogeneous interaction with the ESCRT-subunit tumor susceptibility gene 101 (Tsg101) [117]. The subsequent interaction of ESCRT-1/ESCRT-2 induces endosomal inward budding. The family of charged multivesicular body proteins (CHMPs) then orchestrates the cleavage of the multiple buds, allowing ILV formation [113]. Further, following ILV biogenesis, the MVBs are transported to the cell membrane by the cytoskeletal network, where their fusion with the plasma membrane induces the release of exosomes in the extracellular space. It was shown that the interaction of Rab GTPases and SNARE proteins is a key step in exosomal secretion [118].

#### 4.3.3. Composition of Exosomes in Sjogren’s Syndrome Pathogenesis

The exosomes are formed by different types of proteins, lipids, and nucleic acids. The lipids have been shown to have an important role in cargo sorting to exosomes as well as in cellular signaling. It is important to note that exosome composition is highly dependent on cell type [119]. Throughout the years, multiple studies have demonstrated the presence of specific proteins in exosomes involved in the pathogenesis of SS (Figure 2).

Using salivary gland epithelial cell (SGEC) lines from patients with SS, it was shown that these cells secrete a significantly larger number of exosomes and contain epithelial-specific cytoskeletal proteins for their trafficking. Furthermore, anti-RO/SSA, anti-La/SSB, and Sm ribonucleoproteins were shown to be secreted in exosomes [120]. The anti-RO autoantibodies were shown to be present in up to 70% of patients with SS, whereas an-La autoantibodies have a prevalence of up to 40% [61]. The anti-RO autoantibodies mainly target two RO proteins, Ro52 or Ro60 (also known as SSA). The exosomes could potentially induce SS through the presentation of autoantigens to autoreactive lymphocytes. Another study has shown that the transfer of the EB virus miRNA (i.e., ebv-miR-BART13) to SGECs disturbs salivary secretion by altering calcium entry pathways [121]. In the same way, the transfer of the miR-142-3p microRNA from T-cell exosomes to SGECs impairs salivary secretion by inhibiting the sarcoendoplasmic reticulum Ca^2+^ ATPase 2b (SERCA2B). The downregulation of SERCA2B activity further diminishes cAMP production and Ca^2+^ signaling in SGECs [122]. Moreover, using proteomic analysis, a study identified potential biomarkers in SS patients’ saliva and tears. The biomarkers of the adaptive immune response, the cellular assembly complex, and biomarkers involved in cell metabolism were shown to be upregulated in patients with SS [123]. It was shown that proteins involved in TNF-α-signaling were overexpressed in patients with pSS. Overall, exosomes play an important role in the pathophysiology of pSS by promoting pro-inflammatory pathways.

#### 4.3.4. Exosomes as a Diagnostic Tool for Sjögren’s Syndrome

Novel scientific advances have proposed the use of exosomes as biomarkers in the diagnosis of SS in patients [124]. As mentioned in the previous section, the sensitivity of serological markers (i.e., anti-SSA, anti-SSB, ANA, and RF) in the screening of SS is yet to improve. With the emergence of personalized medicine, the characterization of novel biomarkers in SS could overcome this challenge. Recently, interest in the use of exosomes in the diagnosis of SS has expanded. In this section, the main exosomal cargoes with a potential diagnostic feature in SS are reviewed (Table 1).

In order to identify potential exosomal biomarkers in patients with SS, salivary exosome isolations were performed, followed by RNA extraction and sequencing. The first experiments demonstrated that patients with SS expressed higher levels of miR-23a in their salivary glands, whereas its expression was absent in the exosomes of healthy patients [125]. MicroRNAs are highly studied posttranscriptional regulators of gene expression. The involvement of miR-23a in the pathogenesis of autoimmune disorders has been thoroughly underlined in the literature [126]. In rheumatoid arthritis (RA), miR-23a expression enhancement was shown to be associated with lower levels of inflammation, which can be explained by a downregulation of IL-17 expression and T cell differentiation to Th17 cells. In fibroblast-like synovial cells, miR-23a transfection was shown to inhibit the nuclear factor κ-B kinase subunit α (IKK-α) through TAB2 and TAB3 downregulation. Similarly to RA, miR-23a also showed an anti-inflammatory effect in systemic lupus erythematosus (SLE) and experimental autoimmune encephalitis [127]. Multiple additional microRNAs were found to be upregulated in SS patients with diffuse lymphocytic infiltrates, suggestive of an active disease (see Table 1) [128].

The exosomes show promise as biomarkers that could be used for diagnostic purposes. Kakan et al. successfully identified seven miRNAs that showed significant differences in expression in the lacrimal glands of pSS mouse models [129]. Several studies have also shown that specific biomolecules can be found in exosomes derived from the salivary epithelial cells and T cells of SS patients. In salivary epithelial cells, these potential biomarkers include significant numbers of cytokeratins, highly-expressed EBV miRNA, and intracellular Ro/SSA, LA/SSA, and Sm ribonucleoproteins. In SS patients’ T cells, it was found that exosomes contained a specific miRNA that adversely affects calcium and cAMP pathways, decreasing protein production by salivary epithelial cells. Additionally, exosomes isolated from the tears of SS patients were found to contain more TNF-alpha-related factors and B-cell survival-related proteins [130].

More recently, Yamashiro and his colleagues have established the diagnostic performance of numerous microRNAs for SS [131]. Using mouthrinse samples, they isolated exosomes and RNAs, followed by sequencing and microRNA array assays. Their results showed that nine microRNAs are present in female patients with SS (see Table 1), and they all exhibit various sensitivity and specificity values. The let-7b-5p and miR-34a-5p microRNAs were shown to have the highest specificity values at 83.3% and a sensitivity of 62.5% for both substrates. The most sensitive microRNA in the SS diagnosis was miR-512-3p, with a sensitivity of 87.5% but a specificity of 33.3%. Furthermore, it was shown that combining microRNAs for the diagnosis of SS significantly improves the validity of the test: using the double detection of exosomal microRNAs let-7b-5p and miR-1290 brings the test sensitivity to 91.7% and specificity to 83.3%, with an area under the curve (AUC) value of 0.856 [131]. These results are promising; using exosomes for the diagnosis of SS confers better sensitivity and specificity values in comparison to standard diagnostic biomarkers currently used (see Table 2). Overall, the possibility to screen potential patients for SS by microRNA array assays and sequencing following exosome extraction could allow for faster, non-invasive, and more efficient management. In comparison to serological biomarkers, exosomes are easily extracted from human saliva and are more cost-efficient.

## 5. Conclusions

Sjögren’s syndrome is a chronic autoimmune disease characterized by inflammation of exocrine glands, particularly those involved in tear and saliva production, leading to symptoms such as dry eye. It can also have a variety of systemic complications and present with non-specific symptoms. As of today, the screening and diagnosis of SSDE remain challenges. The current diagnostic criteria for Sjögren’s syndrome dry eye (SSDE) include a combination of clinical tests, blood tests, and evaluation of tear film parameters, but these established diagnostic criteria are either impractical in clinical settings or unreliable. A timely diagnosis of SS is crucial, contributes to improved patient care, and allows monitoring of serious ophthalmic and systemic manifestations.

Novel diagnostic modalities, such as tear and saliva proteomics and exosomal biomarkers, are on the horizon and may improve the accuracy of SSDE diagnosis and allow for earlier referral and treatment, improving the quality of life for patients with Sjögren’s syndrome. Our understanding of the diverse biological processes underlying the pathogenesis of SS will benefit the development of novel biomarkers for early diagnosis to alleviate disease burden. The translation of innovative diagnostic biomarkers from bench to bedside will require extensive clinical trials, but it will present a promising prospect for improving the quality of life of patients suffering from SSDE.

## Figures and Tables

**Figure 1 ijms-24-01580-f001:**
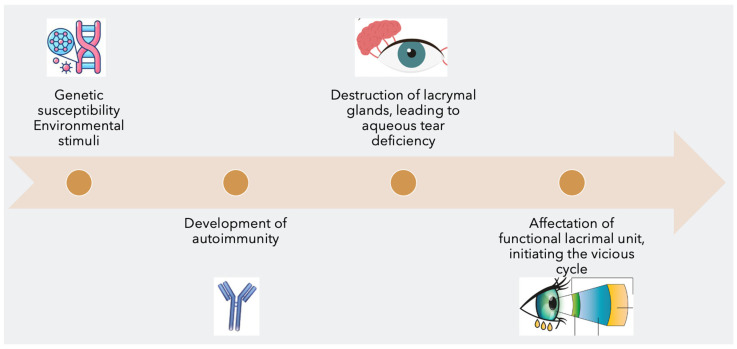
The 4 steps in the pathogenesis of Sjögren’s syndrome dry eye.

**Figure 2 ijms-24-01580-f002:**
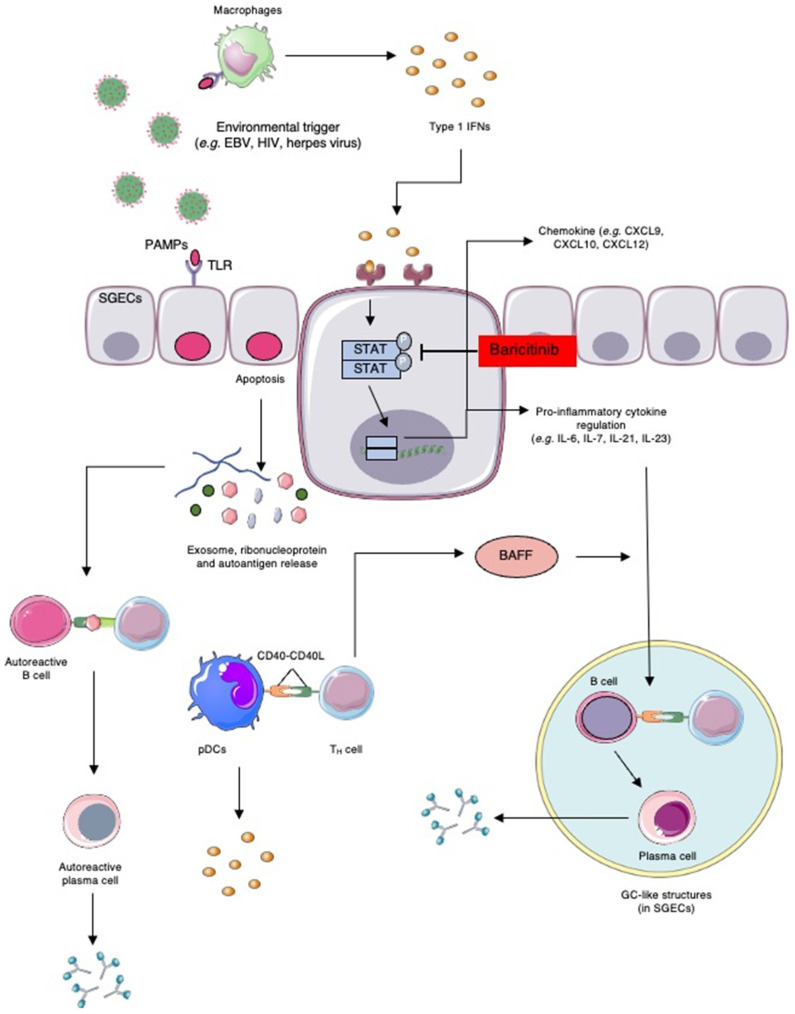
Schematic representation of Sjogren’s syndrome pathogenesis. Environmental triggers, such as viruses, are known to activate innate immunity cells (e.g., macrophages) through pathogen-associated molecular patterns (PAMPs) and pathogen recognition receptors (i.e., toll-like receptors (TLRs)). The activation of TLRs induces a pro-inflammatory microenvironment, marked by the production of type 1 interferons (IFNs). IFNs enhance pro-inflammatory cytokine production by activating the JAK-STAT pathway in salivary gland epithelial cells (SGECs). Ultimately, the pro-inflammating environment leads to the formation of germinal center (GC)-like structures, composed of a tremendous number of activated B and T cells. (The Figure was partly generated using Servier Medical Art, provided by Servier, licensed under a Creative Commons Attribution 3.0 unported license).

**Figure 3 ijms-24-01580-f003:**
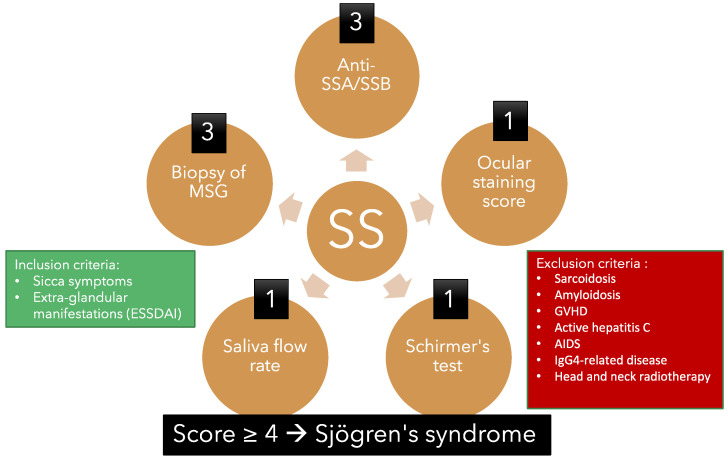
Overview of the American College of Rheumatology/European League Against Rheumatism criteria for primary Sjögren’s syndrome 2016. This classification criteria applies to an individual who meets one of the inclusion criteria and does not have any of the exclusion criteria. A score of 4 or above confirms the diagnosis of pSS.

**Figure 4 ijms-24-01580-f004:**
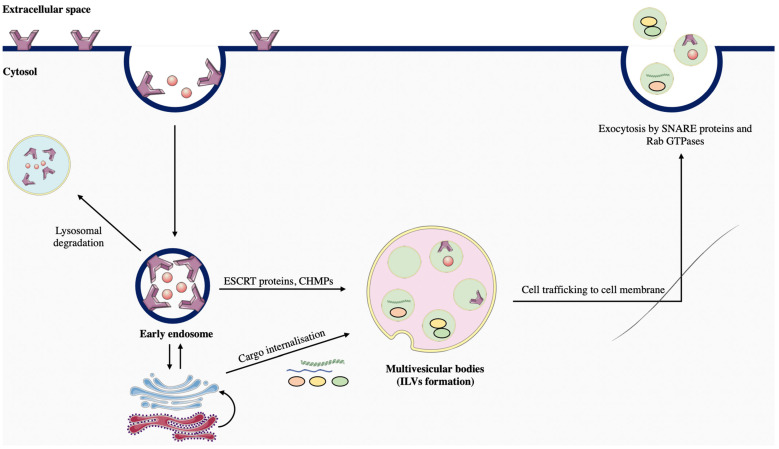
Main steps involved in exosomal biogenesis. Exosome formation is initiated by the endocytic pathway. Endocytosis of cell membrane components and lipids leads to the formation of early endosomes in the cytosol. Endosome sorting by ESCRT and CHMP proteins leads to the formation of multivesicular bodies composed of intraluminal vesicles (ILVs). Cell trafficking processes transfer late endosomes to the cell membrane, where the exocytic pathway expels exosomes in the extracellular space. (The Figure was partly generated using Servier Medical Art, provided by Servier, licensed under a Creative Commons Attribution 3.0 unported license).

**Table 1 ijms-24-01580-t001:** Exosomal biomarkers in Sjogren’s disease and their respective roles.

Biomarker	Role in Sjogren’s Syndrome
Anti-RO/SSA antibodies	Autoantigen presentation to autoreactive lymphocytes
Anti-La/SSB antibodies	
Sm ribonucleoproteins	Pro-inflammatory role
ebv-miR-BART13	Alteration in calcium homeostasis
miR-142-3p	Impaired salivary secretion of salivary grand epithelial cells
miR-23a	Downregulation in Th17 cells and IL-17 levels Inhibition of IKK-α pathway
let-7b-5p	Enhanced expression in SS
miR-1290	
miR-34a-5p	
miR-3648	

**Table 2 ijms-24-01580-t002:** Comparison of the diagnostic performance of exosomal biomarkers.

Biomarker	Sensitivity (%)	Reference
ANA	68.3	Theander et al., 2015 [89]
RF	53	Theander et al., 2015 [89]
Anti-RO/SSA	69–77	Theander et al., 2015 [89]
Anti-La/SSB	39–44	Veenbergen et al., 2022 [90]
let-7b-5p	62.5	Yamashiro et al., 2022 [131]
miR-34a-5p	62.5	Yamashiro et al., 2022 [131]
miR-512-3p	87.5	Yamashiro et al., 2022 [131]
let-7b-5p and miR-1290	91.7	Yamashiro et al., 2022 [131]

## Data Availability

Not applicable.

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
