# Peer review of "An Overview of the Dry Eye Disease in Sjögren’s Syndrome Using Our Current Molecular Understanding"

_ijms, 2023, doi:10.3390/ijms24021580_

Round 1

Reviewer 1 Report

This is a narrative review, so it provides little scientific evidence. However, the authors explain clearly and in detail the characteristics of the disease and the criteria and tools for correct diagnosis.

Author Response

Thank you for your comment. We appreciate your feedback. The objective of our article was to provide a comprehensive understanding of the disease and its diagnosis, with a focus on its molecular pathways and pathogenesis/pathophysiology. As a review article, our aim was to provide an overview and discussion of the current knowledge and understanding of the disease and its diagnosis by using the most recent studies published within the last few years. We hope that this review will be useful for readers seeking to gain a better and updated understanding of the disease and its diagnosis.

Reviewer 2 Report

An interesting article. Other comments and sugestions in the attached document.

kind regards

Author Response

Thank you for your comments and suggestions. We have made the following corrections based on your requests:

  1. Introduction: none
  2. Pathogenesis: none
  3. In the Conventional diagnostic tools and criteria section, we have provided more detailed explanations of how the diagnostic score is calculated and how ocular changes are considered in the diagnosis of Sjogren's syndrome. We have not added a table to present the diagnostic criteria and their associated scores, as the figure 3 already summarize these criteria and the associated scores. 
  4. In the On the Horizon - Novel Diagnostic Modalities section, we have explained the way how these new discoveries can be useful for the diagnosis of Sjogren's syndrome.
  5. In the Conclusions section, we have drawn specific conclusions for each subchapter to provide a more comprehensive summary of the review.

We hope that these changes have addressed your requests and that the revised article is now more clear and informative. Thank you again for your feedback.

Reviewer 3 Report

Please find attachment. Highly valuable work!

Author Response

Thank you for your comments and suggestions. We have made the following corrections based on your requests:

Thank you for your comments and suggestions. We apologize for any errors or omissions in the original version of the article. We have made the following corrections based on your requests:

  1. We have added mention of hormonal changes and stressful events as potential triggers for Sjogren's syndrome.
  2. We have revised the wording to accurately describe the three major components of the tear film, rather than the three layers.
  3. We have emphasized that Sjogren's syndrome has the highest risk for lymphoma development among systemic complications. We have also mentioned that SS has generally a benign clinical course, and provided Harrison's Principal of Internal Medicine as our new reference.
  4. We have added a new section to describe simple questions that can be helpful in considering a diagnosis of Sjogren's syndrome.

We hope that these changes have addressed your requests and that the revised article is now more accurate and informative. Thank you again for your valuable feedback.